



# The second Met Office Unified Model/JULES Regional Atmosphere and Land configuration, RAL2

Mike Bush[1], Ian Boutle[1], John Edwards[1], Anke Finnenkoetter[1], Charmaine Franklin[2], Kirsty Hanley[1], Aravindakshan Jayakumar[3], Huw Lewis[1], Adrian Lock[1], Marion Mittermaier[1], Saji Mohandas[3], Rachel North[1], Aurore Porson[1], Belinda Roux[2], Stuart Webster[1], and Mark Weeks[1]

[1]Met Office, FitzRoy Road, Exeter, EX1 3PB, UK
[2]Bureau of Meteorology (BoM), Melbourne, Victoria, Australia
[3]National Centre for Medium Range Weather Forecasting (NCMRWF), Noida, India

**Correspondence:** Mike Bush (mike.bush@metoffice.gov.uk)

**Abstract.** In this paper we define RAL2 - the second "Regional Atmosphere and Land" (RAL) science configuration for regional modelling. RAL2 uses the Unified Model (UM) as the basis for the atmosphere and the Joint UK Land Environment Simulator (JULES) for the land. RAL2 defines the science configuration of the dynamics and physics schemes of the atmosphere and land and builds on the baseline of RAL1. There are two RAL2 sub-releases, one for mid-latitudes (RAL2-M) and one for tropical regions (RAL2-T). We document the differences between them and where appropriate discuss how RAL2 relates to RAL1 and the corresponding configuration of the global forecasting model. Our results show an increase in medium and low cloud amounts in the mid-latitudes leading to improved cloud forecasts. The increase in cloud amount leads to a reduced diurnal cycle of screen temperature. There is also a reduction in the frequency of heavier precipitation rates. RAL2 is expected to be the last RAL science configuration with two sub-releases as research effort is focused on producing a single defined configuration of the model that performs effectively in all regions of the world.

## 1 Introduction

Regional atmospheric and land models with grid-lengths of the order of a kilometre provide valuable information on local and high-impact weather and are critical to the core function of many National Met Services (NMS) (e.g. Baldauf et al. (2011); Brousseau et al. (2016); Bengtsson et al. (2017); Klasa et al. (2018)).



20    NMSs have to constantly maintain and upgrade their operational systems and make improvements to the skill of their modelling systems in order to fulfil their public service obligations and to demonstrate value for money when investments are made in (for example) high performance (super)computing (HPC). Sometimes these model upgrades will be large and take many years to pull through from research to operations. On other occasions, the upgrades will be more incremental in nature.

The UM partnership consists of a number of institutions that includes the the Met Office, the National Centre for Medium
Range Weather Forecasting (NCMRWF) in India and the Bureau of Meteorology (BoM) in Australia. The regional model is run in areas of interest in different parts of trhe world and it is our goal to have a single defined configuration of the model that performs effectively in all regions.

The Met Office has a Research and Innovation Strategy that sets out aims for the next 10 years and there is an Operational Science Assurance Group (OSAG) that monitors the planned upgrades to operational systems (known as the "Operational
Suite"). The science upgrades are run and validated in a test environment (known as the "Parallel Suite") before being made operational and OSAG signs-off proposed operational changes and determines whether these can be included in an upcoming Parallel Suite. This sign-off process requires results to be presented showing objective verification scores and subjective assessment (carried out with Operational Meteorologists/forecasters), demonstrating the performance of the proposed science changes across a number of standard tests. The computational cost of the proposed change is also an important consideration.

Regional modelling activities at kilometre scale are crucial to the delivery of the Met Office Research and Innovation Strategy and so there is a strong focus on pulling through RAL science configurations into the Operational Suite. In particular, the Operational Suite includes the Met Office's operational deterministic numerical weather prediction (NWP) weather forecast system (the UKV, Tang et al. (2013)) and ensemble prediction system (MOGREPS-UK, Hagelin et al. (2017)). These systems are run with grid-lengths of order of a kilometre and both use the Met Office Unified Model (UM, Brown et al. (2012)) as the
basis for the atmosphere and the Joint UK Land Environment Simulator (JULES, (Best et al., 2011; Clark et al., 2011)) for the land. They are run in variable resolution mode, with horizontal grid-lengths in the central regions of their domains of 1.5km and 2.2km respectively. For climate projection, regional kilometre scale simulations are run with horizontal grid-lengths of 1.5km over a domain covering the Southern UK (Kendon et al. (2014)), 2.2km over Europe (Berthou et al. (2018)) and 4.4km over Africa (Stratton et al. (2018)).

In this paper we define the second "Regional Atmosphere and Land" (RAL) science configuration for kilometre scale modelling using the UM and JULES. "RAL2" defines the science configuration of the dynamics and physics schemes of the atmosphere and land. This configuration has two sub-releases, one for mid-latitudes (RAL2-M) and one for tropical regions (RAL2-T) and builds on the baseline of RAL1 (Bush et al., 2020). Where appropriate, we define how the model configuration relates to the corresponding configuration of the Met Office Unified Model "Global Atmosphere" (GA, Walters et al. (2019)).

In Sect. 2, we document the RAL2 science configuration. In Sect. 3 we evaluate the performance of RAL2-M and RAL2-T configurations in five parts of the world with different meteorology, highlighting the impact of RAL2 developments on performance. Finally, in Sect. 4 we provide some concluding remarks.





## 2 Defining Regional Atmosphere and Land - version 2 (RAL2)

In this section, we give only a brief description of the model, concentrating on the differences from the baseline of RAL1 (Bush et al., 2020), where a more detailed description can be found.

### 2.1 Horizontal and vertical grid

The primary atmospheric prognostics are discretised horizontally onto a longitude/latitude grid. Optionally, this can be a
rotated longitude/latitude grid with the pole rotated so that the grid's equator runs through the centre of the model domain. UK forecasts use this option in order to minimise grid distortion due to convergence of the Meridians, which is most noticeable at high-latitudes. In contrast, domains that are located at lower-latitudes (such as forecasts over India and Australia) use unrotated grids.

    In the vertical, RAL2 uses a 90 level vertical level set labelled $L90(67_t, 23_s)_{40}$, which has 67 levels below $18\,\mathrm{km}$, 23 levels
above this and a fixed model lid $40\,\mathrm{km}$ above sea level. Table 1 compares level sets used in RAL2 and RAL1. The mid-latitude RAL1-M configuration has a 70 level vertical level set labelled $L70(61_t, 9_s)_{40}$, and the tropical RAL1-T configuration has an 80 level vertical level set labelled $L80(59_t, 21_s)_{38.5}$, (Bush et al., 2020).

    The unification of level sets in RAL2 removes an unnecessary difference between mid-latitude and tropical configurations by converging on an enhanced vertical resolution level set that captures the best features of both RAL1 level sets. The
$L70(61_t, 9_s)_{40}$ level set has greater resolution in the lowest 10km of the atmosphere than the $L80(59_t, 21_s)_{38.5}$, whilst the $L80(59_t, 21_s)_{38.5}$ level set has more levels in the upper troposphere than $L70(61_t, 9_s)_{40}$. The rationale for these differences is that the tropopause is shallower in the mid-latitudes than in the tropics. Also, boundary layer fog and low cloud processes are more important in the mid-latitudes and convection more important in the tropics.

    The basic underlying $L90(67_t, 23_s)_{40}$ grid is based on a quadratic function of layer thicknesses, which is gradually stretched
so as to ensure an economical number of levels is employed to cover the height domain of the model. The general method is quite flexible and depends upon the choice of relatively few parameters: the lid top, number of levels, height to which a pure quadratic grid is used, the first layer thickness, a scale parameter for stretching the grid and some simple exponent parameters that govern the rate of stretching (and compression if required).

**Table 1.** Vertical level sets used in RAL1 and RAL2.

| Science Configuration | Number of levels(troposphere;stratosphere) | Height of model top (lid) |
|---|:---:|---:|
| RAL1-M | L70(61t;9s) | $40\,\mathrm{km}$ |
| RAL1-T | L80(59t;21s) | $38.5\,\mathrm{km}$ |
| RAL2 | L90(67t;23s) | $40\,\mathrm{km}$ |



## 2.2 Dynamical core: Spatio-temporal discretisation

The ENDGame dynamical core is a semi-implicit (SI) semi-Lagrangian (SL) formulation that solves the non-hydrostatic, fully-compressible deep-atmosphere equations of motion (Wood et al., 2014).

## 2.3 Lateral Boundary Conditions (LBCs)

The treatment of LBCs uses the method of relaxation/blending (Davies, 1976; Perkey and Kreitzberg, 1976).

## 2.4 Solar and terrestrial radiation

The SOCRATES [1], (last access: 18 September 2022) radiative transfer scheme (Edwards and Slingo, 1996; Manners et al., 2018) is used with a configuration based on GA3.1 (Walters et al., 2011). Solar radiation is treated in six shortwave bands and thermal radiation in nine longwave bands.

## 2.5 Microphysics

A single moment microphysics scheme based on Wilson and Ballard (1999), but extensively modified is used. Prognostic rain and prognostic graupel are included. The warm-rain scheme is based on Boutle et al. (2014b) whilst ice cloud parametrisations use the generic size distribution of Field et al. (2007) and mass-diameter relations of Cotton et al. (2013).

## 2.6 Large-scale cloud

RAL2-M uses the Smith (1990) cloud scheme. This is a diagnostic scheme which relies on a definition of critical relative
humidity, RHcrit, the grid-box mean relative humidity at which clouds start to appear. For liquid cloud, the Smith cloud scheme is built around an assumption that sub-grid temperature and humidity fluctuations can be described by a symmetric triangular probability distribution function (PDF). An empirically-adjusted cloud fraction (EACF) and an area cloud fraction scheme are also used, which follows a similar approach to that described by Boutle and Morcrette (2010).

The ice cloud fraction is parametrised as described by Abel et al. (2017) where it is diagnosed from the ice water con-
tent. A change at RAL2-M is to limit the overlap between the liquid water and ice phases. Abel et al. (2017) describe how aircraft observations in a cold-air outbreak to the north of the United Kingdom are used to examine the boundary layer and cloud properties in an overcast mixed-phase stratocumulus cloud layer and across the transition to more broken open-cellular convection. Sensitivity studies using a convection-permitting (1.5-km grid spacing) regional version of the Met Office Unified Model showed that ice was too active at removing supercooled liquid water from the cloud layer and that improvements could
be made by limiting the overlap between the liquid water and ice phases. Reducing the ice cloud fraction in mixed-phase regions protects a region of supercooled liquid and prevents excessive depletion of this by riming. This delays the transition of cold-air outbreaks into snow showers and improves the reflected SW radiation by increasing stratiform regions.

---

[1]https://code.metoffice.gov.uk/trac/socrates





RAL2-T has three extra prognostic fields (liquid fraction, ice fraction and mixed-phase fraction) as it uses the prognostic cloud prognostic condensate (PC2) cloud scheme (Wilson et al., 2008a).

## 2.7 Atmospheric boundary layer

Although most turbulent motions are still unresolved in kilometre scale models, the largest scales can be of a similar size to the grid-length. The model must therefore be able to parametrize the smaller scales, resolve the largest ones if possible, and not
alias turbulent motions smaller than the grid-scale onto the grid-scale. The "blended" boundary-layer parametrisation described by Boutle et al. (2014b) is used to achieve this. This scheme transitions from the 1D vertical turbulent mixing scheme of Lock et al. (2000), suitable for low-resolution simulations such as GA configurations, to a 3D turbulent mixing scheme based on Smagorinsky (1963) and suitable for high-resolution simulations, based on the ratio of the grid-length to a turbulent length scale. The blended eddy diffusivity, including any non-local contribution from the Lock et al. (2000) scheme, is applied to
down-gradient mixing in all three dimensions, whilst appropriately weighted non-local fluxes of heat and momentum are retained in the vertical for unstable boundary-layers.

A change that is included in RAL2 is the addition of the "Leonard" and "Cross" terms as proposed by Moeng et al. (2010). These terms are part of the vertical turbulent flux and are parameterised as a function of local horizontal gradients. The implementation of the Leonard terms in the UM is described in Hanley et al. (2019). RAL2 also includes a number of minor
corrections to the Smagorinsky scheme, including the horizontal diffusion of cloud liquid water and the use of the momentum diffusion coefficient to diffuse vertical velocity in the vertical.

The configuration of the Lock et al. (2000) scheme is the same as that of GA7 (Walters et al., 2019), except for the following differences: (i) for stable boundary layers, the "sharp" function is used everywhere, but with a parametrisation of sub-grid drainage flows dependent on the sub-grid orography (Lock, 2012), (ii) heating generated by frictional dissipation of turbulence
is not represented, (iii) the parametrisation of shear generated turbulence extending into cumulus layers (Bodas-Salcedo et al., 2012) is not used and (iv) RAL2 to use the surface fluxes calculated by JULES, rather than a simpler, less accurate calculation used in RAL1.

**Table 2.** RAL2-M and RAL2-T differences. %

| Science difference | RAL2-M | RAL2-T |
|---|---|---|
| BL Free Atmospheric mixing length | 40m | interactive mixing length |
| BL Stability functions | $b_{LEM}$ =1.43, $c_{LEM}$ =1.43 | $b_{LEM}$ =40, $c_{LEM}$ =16 |
| BL stochastic perturbations to temperature and moisture | on (improved triggering) | off |
| Cloud Scheme | Smith (diagnostic) | PC2 (prognostic) |

There are two differences in the representation of turbulence between RAL2-M and RAL2-T, namely in the form of the unstable stability functions and in the free-atmospheric mixing length. Both give enhanced turbulent mixing in RAL2-T com-





pared to RAL2-M. RAL2-M uses the Brown (1999) "conventional" function, the same as GA7, while RAL2-T uses the Brown (1999) "standard" function. RAL2-T has an interactive free-atmospheric mixing length, whilst RAL2-M uses a value of 40m. Related to this, stochastic perturbations to temperature and specific humidity are applied to RAL2-M (but not RAL2-T) in an effort to improve the triggering of explicit convection as described for RAL1 in Bush et al (2020). For more details and a summary of differences between RAL2-T and RAL2-M, see Table 2.

## 2.8   Land surface and hydrology

The community land surface model JULES (Best et al., 2011; Clark et al., 2011) represents exchanges of mass, momentum and energy between the atmosphere and the underlying land and sea surfaces. The configuration adopted in RAL2 largely follows that of GL7.0 (Walters et al., 2019), although different priorities for regional and global modelling development can result in differences between the configurations.

A fixed value of Charnock's coefficient (0.011) is used to determine the surface roughness over open sea. Parametrisation of the sea surface albedo is based on Barker and Li (1995) and an RAL2 change implements form drag over sea ice bringing the treatment up to the level of GL8.0 (no reference). RAL2 also limits drag over the ocean at high wind speeds by imposing a cap on the drag coefficient in very high winds. This is more realistic than allowing the drag coefficient to increase continually and significantly improves the wind-pressure relationship of tropical cyclones.

To include the effect of convective boundary layer and cloud-scale gusts on the surface turbulent fluxes, we set

$$v_*^2 = u_*^2 + \beta^2 w_*^2 + \Gamma^2 w_c^2 \qquad (1)$$

where $u_*$ is the friction velocity from the mean wind, $w_*$ the PBL convective velocity scale and $w_c$ gustiness from parametrized convective downdraughts.

$\beta$ is reduced from 0.08 to 0.04 in RAL2 to improve consistency with GL7.0 (which follows Redelsperger et al. (2000), where

0.04 gave better agreement with LES of simple PBL turbulence) thus reducing the convective gustiness contribution to surface exchange.

RAL2 includes the multilayer snow scheme, with a value for the density of fresh snow of 170 $\mathrm{kgm^{-3}}$. Improvements to the treatment of lying snow in RAL2 are achieved by introducing a representation of melting of the snow pack from the base over warm ground, as the original code in JULES allows melting only from the surface. This modification allows the reintroduction

of graupel into the precipitation reaching the surface. A small bug-fix to the calculation of the albedo of thin snow is included. The treatment of the growth of snow grains (introduced into the global model at GL8.0) is also revised.

In GL7.0 urban surfaces are represented by a single urban tile, but in RAL2 two separate tiles for street canyons and roofs are used for UK domains (Porson et al., 2010). Currently the two tile scheme is limited to domains over the UK due to the availability of morphology data.





### 2.9 Lower boundary condition (ancillary files) and forcing data

In the UM, the characteristics of the lower boundary, the values of climatological fields and the distribution of natural and anthropogenic emissions are specified using ancillary files. Table A1 in the appendix contains the main ancillaries used in RAL applications as well as references to the source data from which they are created.

## 3 Model evaluation

In this section we demonstrate the performance of RAL2 comparing to the baseline of RAL1. A range of evaluation methods are required to assess the performance. Verification skill scores, diagnostic plots and case studies all provide useful information on model characteristics and skill. The regional model evaluation process benefits from the multi-institutional UM partnership and the regional model is run by UM partners in a variety of domains worldwide. We have focused on performance of RAL2 over the UK, Australia and India. This allows us to assess the model behaviour in a variety of climatic zones and for different weather phenomena. We give only a brief description of the evaluation metrics and the Regional Model Evaluation and Development (RMED) Toolbox, as a more detailed description can be found in Bush et al. (2020).

The "High Resolution Assessment" (HiRA) framework (Mittermaier, 2014) provides a spatial and inherently probabilistic framework for evaluating kilometre scale models. HiRA uses synoptic observations and a neighbourhood of model grid points centered on observation locations. The HiRA Continuous Ranked Probability Score (CRPS) is used for temperature and the Ranked Probability Score (RPS) is used for non-normally distributed or spatially discrete variables such as precipitation.

Precipitation is also evaluated using the Fractions Skill Score (FSS, Roberts and Lean, 2008). The FSS requires a spatial observation-based analysis and over the UK this is a radar-based analysis, whilst in the tropics (for example in South-East Asia) a GPM based product (Skofronick-Jackson et al., 2017) is used.

### 3.1 The Regional Model Evaluation and Development (RMED) Toolbox

The main purpose of the RMED toolbox is to ensure a uniformity of verification and diagnostic output across multiple users and institutions. One of the outputs of the toolbox is a 'scorecard' - a single clear plot with arrows/triangles showing whether the model version being tested is better or worse than a previous incarnation. Triangles pointing upward (green) indicate that the test model is better than the control and downward (purple) triangles indicate the control model is better. The area of the triangles is proportional to the absolute improvement (or deterioration) of the model and the triangles are outlined in black if the change is statistically significant at the 0.05 level determined using the Wilcoxon signed-rank test. The scorecards contain a huge amount of information, digested into an easy-to-understand summary, allowing fast assessments about model skill to be made. Other outputs include domain (area) average plots , histograms and "cell statistics" (Hanley et al., 2015).





## 3.2 Performance of individual science changes

In this section we illustrate the impact of the RAL2 changes on model performance. The baseline used for the UK and mid-latitudes is RAL1-M. Individual science changes (see list of RMED tickets in Table A3) were tested by running 100 case studies with a 1.5km horizontal grid-length, using the same domain as the Operational UKV model (Figure 1). The cases were simple downscaling runs (from the Met Office Global model) with no data assimilation. The cases sampled a wide range of
meteorological conditions from the period July 2014 to April 2017 and comprised roughly equal numbers from each season. The cases were a mixture of poor forecasts (as identified by forecasters), high impact weather and normal everyday weather.

Case studies were also run for a domain over Darwin, Australia in order to assess performance in the tropics. Darwin is the preferred location for tropical testing as there are observations from the Darwin C-band polarimetric radar which collects 3D observations out to a range of 150 km (Louf et al., 2018), which allows for a detailed evaluation of simulated tropical
convection. Figure 2 shows the domain the radar covers and the area over which the comparison with the model is done.

Figure 3 shows a case study from 18th November 2016 in which a thin layer of graupel over SW England in the Operational UKV forecast (RAL1) motivated an emergency change to remove graupel at the surface being seen by JULES. Ticket 20 (improvements to the treatment of lying snow, see section 2.8) includes graupel and applies existing code for melting below needle-leaved trees, instantaneously melting if soil is above freezing. It removes the spurious very thin snow shown in RAL1,
leading to a warming in those areas (e.g. over Ireland, Wales, South-West England and Northern France). Figure 4 shows scorecard verification for Ticket 20 with screen temperature and visibility showing statistically significant improvements.

Figure 5 shows scorecard verification for ticket 27 (Leonard Terms, see section 2.7). The overall impact is neutral, with a slight improvement to cloud base height and a slight detriment to visibility. The top panel in Figure 6 shows the histogram of rain rates and shows a reduction in high rates above 10mm/hr. The middle panel in Figure 6 shows the frequency of occurrence
of precipitation in a convective cell. The frequency is reduced for all rates and this is in closer agreement with GPM obs for lower rates, but worse agreement at higher rates. The bottom panel in Figure 6 shows there are fewer small cells, showing better agreement with GPM obs.

Figure 7 shows scorecard verification for ticket 38 (Improved ice cloud fraction in mixed phase clouds, see section 2.6). There is a detriment to screen temperature and improvement to cloud fraction, visibility and precipitation. Figure 8 shows
medium and low cloud amounts are increased.

Scorecard verification for tickets 30, 36, 37, 39, 42 and 43 was neutral, showing no statistically significant changes in performance either over the UK or the Darwin domains. Tickets 20, 27 and 38 had a positive impact over the UK and neutral impact over Darwin, allowing the decision to be taken to combine the tickets togther into a package of changes referred to as RAL2-M in the mid-latitudes and RAL2-T in the tropics.

## 3.3 Mid-Latitude performance of RAL2 case studies over the UK

Figure 9 shows scorecard verification for RAL2-M vs RAL1-M. There is improvement to all variables with statistically significant results at 7 grid lengths for temperature, cloud (fraction and base), visibility and precipitation. Stratifying the cases





by season reveals that the performance in Winter (Figure 9 middle panel) is much better than the performance in Summer (Figure 9 bottom panel).

The signals noted in section 3.2 are also seen in the RAL2 case studies, with an increase in medium and low cloud and decreased precipitation amounts associated with a reduction in the the frequency of heavier rates (not shown).

### 3.4 Mid-Latitude performance of RAL2 Data Assimilation trials over the UK

RAL2 was tested with the operational 4DVAR data assimilation system (Milan et al., 2020) in use at the time (known as Operational Suite 42, OS42). This was operational from 19th March 2019 to 04th December 2019. It was decided that RAL2 would be aimed at the next Parallel Suite (known as Parallel Suite 43, PS43), which would eventually become operational on 04th December 2019 (and be known as Operational Suite 43, OS43). It was also decided that despite RAL2 being defined as using a $L90(67_t, 23_s)_{40}$ level set, the implementation of RAL2 in the Parallel Suite would retain the $L70(61_t, 9_s)_{40}$ level set

due to the extra cost of the $L90(67_t, 23_s)_{40}$ level set.

The UKV 4D-VAR Winter trial was run for thirty-eight days of the Winter 2017 period (01/12/2017 - 08/01/2018) and eight weeks of the Summer 2018 period (15/07/2018 - 18/08/2018). Figure 10 shows the bias for screen temperature and cloud amount vs lead time for 00 UTC, 06 UTC, 12 UTC and 18 UTC runs in Summer. There is a good correlation between the cooler temperatures by day in RAL2 (which verifies worse) and the increased cloud cover (which verifies better, reducing a

negative bias).

### 3.5 Mid-Latitude performance of RAL2 MOGREPS-UK trials over the UK

At OS41, the MOGREPS-UK ensemble system is a 6 hour cycling, 12 member ensemble driven by MOGREPS-G LBCs and centred around the UKV analysis. Initial condition uncertainty is sampled by adding perturbations from MOGREPS-G members and forecast uncertainty is sampled by the random parameter (RP) scheme (McCabe et al., 2016) to perturb the

model physics. At OS42 the MOGREPS-UK system moved to an hourly cycling system and although both OS41 and OS42 MOGREPS-UK trials have been run with RAL2, only results from the OS41 runs are shown.

The MOGREPS-UK trials were run for one month in Summer 2017 (02/07/2017 to 02/08/17) and one month in Winter 2017-2018 (02/12/2017 to 02/01/2018). Figure 11 shows RAL2 to outperform RAL1 in Winter with improvements to screen temperature, cloud base height, visibility and precipitation. There is a detriment to wind which is statistically significant at a

number of forecast ranges. There is a detriment to cloud fraction at early forecast ranges to T+4, but an improvement from T+12 onwards. In Summer RAL2 also outperforms RAL1 with improvements to cloud fraction, cloud base height, visibility and precipitation. There is a detriment to screen temperature whilst wind shows a neutral signal.

The MOGREPS-UK verification results are consistent with the results from the case studies (section 3.3) and the UKV DA trials (section 3.4) with better temperature performance in Winter compared to Summer (Figure 9 middle vs bottom panel),

due to the aforementioned worsened daytime cold bias in Summer (Figure 10).





### 3.6 Mid-Latitude performance - Perth (Australia) fog case

The Australian evaluation was carried out at BoM in Australia and consisted of running 8 case studies over various domains with a 1.5km horizontal grid-length. Here and in the next section, we discuss two of the 8 cases.

Fog was observed at Perth Airport between 1600-2300 UTC on 29/08/17 (1am-7am local on 30/08/17). The tropical configurations RAL1-T and RAL2-T have more extensive fog than mid-latitude configurations RAL1-M and RAL2-M with little
difference between RAL1 and RAL2 (not shown). It should be noted that some parameters in the visibility diagnostic have been tuned at BoM to better suit fog conditions in Australia. Mid-latitude configurations have more low cloud and less high cloud and are warmer and drier than tropical configurations though the evening transition and nighttime minimum.

### 3.7 Tropical performance - Darwin MCS case

The case studied is the 18th of February 2014 where active monsoon conditions produced a mesoscale convective system
(MCS). The observed and modelled MCS lifecycle is illustrated in Figure 12, which shows the fractional area of the radar domain covered by reflectivities greater than 10 dBZ as a function of height and time over a 12-hour period. The observations come from the Darwin C-band polarimetric radar which collects 3D observations out to a range of 150 km (Louf et al., 2018), which allows for a detailed evaluation of simulated tropical convection. (Figure 2 shows the domain the radar covers and the area over which the comparison with the model is done.)

From 12 - 15 UTC scattered convection was observed around Darwin, and by 17UTC the convection had become organised. Throughout this time, all the configurations produce too much cloud cover, deeper clouds and more rainfall in the domain than was observed by the radar. The largest difference between RAL2-M and RAL2-T is the greater area covered by cloud and rain in the RAL2-T simulation from 18UTC. This corresponds to the time when the MCS matured and had an extensive stratiform cloud region. The largest fractional areal coverage is 0.9 in the RAL2-T simulation, which agrees with the observed value albeit
the simulated maximum is a couple of hours too early. Compared to RAL1-T, RAL2-T shows improvements in the longer time where large fractional coverage >0.8 was observed, as well as the larger areal coverage of rain below the melting level.

### 3.8 Tropical performance - South East Asia cases

The South East Asia evaluation was carried out as part of a WCSSP South East Asia project at the Met Office and consisted of near real time running of a 4.4km horizontal grid-length model covering a large domain covering Indonesia, Singapore,
Malaysia and the Philippines. The model was run twice per day from 20th November 2018 through to 17th December 2018 giving a total of 56 cases. Figure 13 shows RAL2-T outperforms RAL1-T with improvements to screen temperature, cloud fraction, cloud base height and wind. FSS results (Figure 13 bottom panel) show a significant improvement to precipitation at all thresholds between T+24 and T+72. The only degradation is seen in the first few hours of the forecast when convective-scale structures are still spinning up from Global model initial fields at T+0.





### 3.9 Tropical performance - Two Indian lightning cases

The Indian evaluation was carried out at NCMRWF in India and consisted of a number of case study runs with a 4.0km horizontal grid-length model covering all India and looking primarily at rainfall and lightning. Here, we discuss two fairly intense lightning cases. The lightning flash counts by RAL2 were underestimated compared to RAL1 and hence a tuning was carried out by reducing the GWP threshold for the storm detection from 200 gm-2 to 100 gm-2 which has enhanced the flash counts at par with RAL1 values.

The cases studied are (i) 02nd May 2018 where widespread lightning occurred associated with an MCS over the Northern sector of Indian Great Plains and (ii) 16 April 2019, a case of strong Western Disturbance causing widespread rainfall over north-central and north-west India. The first case did not have enough coverage of observations over the entire Indian region to verify except Chinese satellite FY-4A LMI covering only the Eastern sector of India. Whilst the second case has lightning observations coming from two sources, ie., Indian Institute of Tropical Meteorology (IITM) and Indian Air Force (IAF) ENLS datasets which are merged and binned at 4km resolution.

Figure 14 shows that RAL2-T (middle row) has slightly fewer total lightning flashes compared to RAL1-T (upper row). This is due to a reduction in both the graupel water path and ice water path. Although the vertical velocity (updraft) in RAL2 is higher than RAL1 over some pockets, this appears to be of secondary importance to graupel/QCF amount on the flash rate. The second case shows fairly good match between the model (Figure 15 upper and middle rows) and observations (Figure 15 bottom panel) for both RAL1 and RAL2. The observations show very few intense hotspots for the second case with the counts extending even up to 50 (over the foothills of Bihar and Uttar Pradesh and also very few flash strikes over the Rajasthan-Madhya Pradesh border). RAL1 and RAL2 both show a fairly good match over the Himalayan region but with a slight shift towards the upper slopes, while the central Indian hotspots are missing in both simulations. The maximum counts are reduced from 40 in RAL1 to 30 in RAL2 with both simulations showing too large a coverage of the intense patch compared to observations.

### 4 Conclusions

In this paper we have defined the RAL2 Science configuration of the regional Met Office Unified model. RAL2 is an important step in the development of kilometre grid scale configurations of the Unified Model and we define two sub-releases, one for mid-latitudes (RAL2-M) and one for tropical regions (RAL2-T). Results are presented from simulations both with and without Data Assimilation (the latter we refer to as "case studies"), from deterministic and ensemble runs and from domains in both the mid-latitudes (U.K and Perth in Australia) and the tropics (Darwin in Australia, South East Asia and India).

The recent science developments included in RAL2-M are shown to increase medium and low cloud amounts and decrease precipitation amounts (associated with a reduction in the frequency of heavier rates). The diurnal cycle of temperature sees a warming compared to RAL1 from early evening through the night time period in Winter, reducing a cold bias at this time. In Summer, there is a reduction in maximum temperature in RAL2-M compared to RAL1-M which worsens the cold bias. These temperature changes (warmer by night and cooler by day) are consistent with the increased cloud cover, which verifies better, reducing a negative bias. Visibility forecasts over the U.K in Winter are improved, although the simulation of a fog case at





Perth Airport in Australia showed similar performance to RAL1. There is a consistency in performance between individual science change tests (section 3.2), RAL2 case studies (section 3.3), data assimilation trials (section 3.4) and MOGREPS-UK trials (section 3.5).

RAL2-T outperforms RAL1-T in the South East Asia region of the tropics with significant improvement to precipitation at all thresholds between T+24 and T+72. There are also improvements to screen temperature, cloud fraction, cloud base height
and wind. Results from other tropical tests over Darwin, Australia and India show incremental changes to model behaviour.

At the Met Office, RAL2 was implemented Operationally at Parallel Suite 43 (PS43) and since 04th December 2019 (and to this day), RAL2-M science is running 24/7 in the UKV and MOGREPS-UK weather forecast systems. Despite RAL2 being defined as using a $L90(67_t, 23_s)_{40}$ level set, the implementation of RAL2 in Operational weather forecasting retains the $L70(61_t, 9_s)_{40}$ level set due to cost. It is currently planned to upgrade to the $L90(67_t, 23_s)_{40}$ level set in 2023 as part of the
exploitation of a new HPC.

Looking ahead to RAL3, research effort is focused on producing a single defined configuration of the model that performs effectively in all regions of the world. This goal is hugely challenging and will require a concerted effort and coordination from the partnership developing the RAL configuration. In this paper we have shown a series of tests in a small number of regions that requires substantial computational effort. For RAL3, we will need to develop a more extensive set of tests for the
model that gives confidence that changes are generally improving the system. One very specific area which is not covered in this paper is the performance of the model in climate simulations. It remains a high priority to include climate testing in the development process of the regional model although with the high computing costs involved in regional climate runs at the kilometre gridscale system, the test will need careful design.

*Code availability.*

Due to intellectual property right restrictions, we cannot provide the source code or documentation papers for the UM.

*Obtaining the UM.* The Met Office Unified Model (UM) is available for use under a close licence agreement. A number of research organizations and national meteorological services use the UM in collaboration with the Met Office to undertake research, produce forecasts, develop the UM code, and build and evaluate models. For further information on how to apply for a licence, please get in contact with scientific_partnerships@metoffice.gov.uk or see http://www.metoffice.gov.uk/research/
modelling-systems/unified-model (last access: 19 September 2022). UM documentation papers are accessible to registered users at https://code.metoffice.gov.uk/doc/um/latest/umdp.html.

*Obtaining JULES.* The JULES user manual is accessible via https://jules-lsm.github.io/ and JULES is available under licence free of charge. For further information on how to gain permission to use JULES for research purposes see http://jules-lsm.github.io/access_req/JULES_access.html (last access: 19 September 2022).

*Details of the simulations performed.* UM/JULES simulations are compiled and run in suites developed using the Rose suite engine (http://metomi.github.io/rose/doc/html/index.html, Met Office, 2022) and scheduled using the cylc workflow engine (https://cylc.github.io/cylc, Oliver et al. (2019)). Both Rose and cylc are available under v3 of the GNU General Public License





(GPL). In this framework, the suite contains the information required to extract and build the code as well as configure and run the simulations. Each suite is labelled with a unique identifier and is held in the same revision-controlled repository service in which we hold and develop the model code. Therefore these suites are available to any licensed user of both the UM and JULES.

*Obtaining FCM.* The UM and JULES codes were built using the fcm_make extract and build system provided within the
5  Flexible Configuration Management (FCM) tools. UM and JULES codes and Rose suites were also configuration managed using this system. Further information is provided at http://metomi.github.io/fcm/doc/user_guide/ (last access: 19 September 2022). We document a set of reference RAL2-based simulations in Table 3.

**Table 3.** Identifiers for a set of RAL2 reference simulations across a number of systems/applications. These suites are held on the Met Office Science Repository Service, which also holds the UM and JULES code. %

| Application | Suite id | UM version/JULES version |
|---|---|---|
| UKV case studies | u-bc363 | UM11.1/JULES5.2 |
| UKV 4D VAR trial suite | mi-ay695 and mi-ay697 | UM11.1/JULES5.2 |
| MOGREPS-UK case studies | mi-ay685_win and mi-ay685 | UM11.1/JULES5.2 |
| Perth fog case study | u-av356 | UM11.1/JULES5.2 |
| Darwin MCS case study | u-av356 | UM11.1/JULES5.2 |
| South East Asia case studies | u-av356 | UM11.1/JULES5.2 |
| India lightning case studies | u-av356 | UM11.1/JULES5.2 |





## Appendix A

### A1

We list source datasets used to create standard ancillary files used in RAL2 in Table A1.

**Table A1.** Source datasets used to create standard ancillary files used in RAL2. [%]

| Ancillary field | Source data | Notes |
|---|---|---|
| Land Sea mask | IGBP; Loveland et al. (2000) | Used for UKV/MOGREPS-UK. |
| | CCI; Hartley et al. (2017) | CCI mask lacking in inland lakes definition |
| Mean/sub-grid orography | DTED 1km ; | Used for UKV/MOGREPS-UK. |
| | GLOBE 30″; Hastings et al. (1999) | Fields filtered before use |
| | SRTM; Bunce et al. (1996) | Shuttle Radar Topography Mission. Mean orography only. |
| | | Available up to 60 degrees North. |
| Land usage | IGBP; Loveland et al. (2000) | Mapped to 9 tile types |
| | ITE; Bunce et al. (1996) | U.K. only |
| | CCI; Hartley et al. (2017) | European Space Agency Land Cover Climate Change Initiative |
| Soil properties | HWSD; Nachtergaele et al. (2008) | Three datasets blended via optimal interpolation |
| | STATSGO; Miller and White (1998) | |
| | ISRIC-WISE; Batjes (2009) | |
| Leaf area index | MODIS collection 5 | 4 km data (Samanta et al., 2012) mapped to 5 plant types |
| Plant canopy height | IGBP; Loveland et al. (2000) | Derived from land usage and mapped to 5 plant types |
| Bare soil albedo | MODIS; Houldcroft et al. (2008) | |
| SST/sea ice | System/experiment dependent | |
| Ozone | Li and Shine (1995) | |
| Murk aerosol | NAEI, ENTEC and EMEP emission inventories | |
| CLASSIC aerosol climatologies | System/experiment dependent | Used when prognostic fields not available |





## A2

We list acronyms in Table A2.

**Table A2.** Acronym list. %

| Acronym | Meaning | Notes |
|---------|---------|-------|
| EACF | Empirically Adjusted Cloud Fraction | |
| ENDGame | Even Newer Dynamics for General atmospheric modelling of the environment | Dynamical core |
| GA | Global Atmosphere | Global Atmosphere science configuration |
| GA3.1 | Global Atmosphere 3.1 | A specific GA science configuration |
| GA7.0 | Global Atmosphere 7.0 | A specific GA science configuration |
| GL | Global Land | Global Land science configuration |
| GL7.0 | Global Land 7.0 | A specific GL science configuration |
| JULES | Joint UK Land Environment Simulator | Community Land surface model |
| LAM | Limited Area Model | |
| LBCs | Lateral Boundary Conditions | |
| MOGREPS-UK | Met Office Global and Regional Ensemble system - UK | UK NWP operational ensemble system |
| NMS | National Met Services | |
| NWP | Numerical Weather Prediction | |
| RAL | Regional Atmosphere and Land | |
| RAL1 | Regional Atmosphere and Land 1 | First RAL science configuration |
| RAL1-M | Regional Atmosphere and Land 1 - Mid Latitudes | |
| RAL1-T | Regional Atmosphere and Land 1 - Tropics | |
| RAL2 | Regional Atmosphere and Land 2 | Second RAL science configuration |
| RAL2-M | Regional Atmosphere and Land 2 - Mid Latitudes | |
| RAL2-T | Regional Atmosphere and Land 2 - Tropics | |
| RMED | Regional Model Evaluation and Development | |
| SOCRATES | Suite Of Community RAdiative Transfer codes based on Edwards and Slingo | Radiative Transfer scheme |
| UKV | UK Variable (resolution) | UK NWP operational deterministic model |
| UM | Unified Model | |



## A3

The Regional Model Evaluation and Development (RMED) processes at the Met Office makes use of an online 'ticket' tracking system which allows scientists to document changes to the model. RMED tickets included in RAL2 are listed in Table A3. These are the RAL2 developments which when added to the RAL1 base define RAL2. The developments are ordered by ticket number to both inform the development community and for future cross-reference.

**Table A3.** RMED tickets included in RAL2. %

| RMED Ticket number | RAL2-M/RAL2-T | Description of RAL2 change |
|---|---|---|
| 20 | RAL2-M and RAL2-T | Improvements to the Treatment of Lying Snow |
| 27 | RAL2-M and RAL2-T | Leonard Terms |
| 30 | RAL2-M and RAL2-T | Minor corrections to the Smagorinsky scheme, including horizontal diffusion of liquid cloud |
| 36 | RAL2-M and RAL2-T | Unify vertical level sets in Mid-Latitude and tropical configurations |
| 37 | RAL2-M and RAL2-T | Implement form drag over sea ice |
| 38 | RAL2-M | Improved ice cloud fraction in mixed phase clouds |
| 39 | RAL2-M and RAL2-T | Use real surface fluxes in convection diagnosis |
| 42 | RAL2-M and RAL2-T | Reduce convective gustiness contribution to surface exchange to be consistent with GA |
| 43 | RAL2-M and RAL2-T | Limit drag over the ocean at high wind speeds |

5  *Author contributions.*

Mike Bush led the RAL2 testing and evaluation process and prepared the manuscript with contributions from all co-authors. Ian Boutle, John Edwards, Kirsty Hanley and Adrian Lock are either code owners and/or developers of the model code included in RAL2. Ian Boutle, John Edwards, Anke Finnenkoetter, Charmaine Franklin, Kirsty Hanley, Jayakumar, Adrian Lock, Saji Mohandas, Aurore Porson, Belinda Roux, Stuart Webster and Mark Weeks performed the evaluation. Marion Mittermaier and
10  Rachel North contributed to the writing of the model evaluation section.

*Competing interests.*

The authors declare that they have no conflict of interest.

*Acknowledgements.* The development and assessment of the Regional Atmosphere Land configurations is possible only through the hard work of a large number of people that exceeds the list of authors. Specifically we would like to thank Chris Short [1] for his work with the





15    RMED Toolbox and Vinod Kumar for his efforts in installing/supporting the RMED Toolbox/RES at the Bureau [2].

[1] Met Office, Exeter, UK

[2] Bureau of Meteorology (BoM), Melbourne, Victoria, Australia

[3] National Centre for Medium Range Weather Forecasting (NCMRWF), Noida, India

The research/project work of research and project work of participants from the Bureau of Meteorology was undertaken with assistance of

5    resources and services from the National Computational Infrastructure (NCI), which is supported by the Australian Government.

The GPM IMERG Late Precipitation L3 Half Hourly 0.1 degree x 0.1 degree V04 data were provided by the NASA/Goddard Space Flight Center's Goddard Earth Sciences Data and Information Services Center and PPS, which develop and compute the GPM IMERG Late Precipitation L3 Half Hourly 0.1 degree x 0.1 degree as a contribution to GPM, and archived at the NASA GES DISC.





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

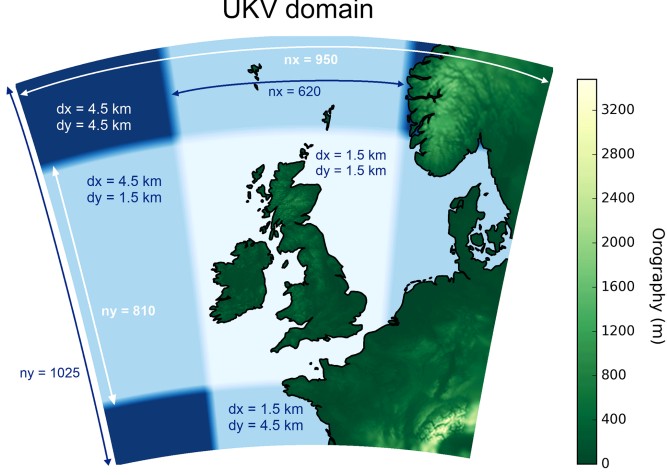

**Figure 1.** Domain for UK Case studies.





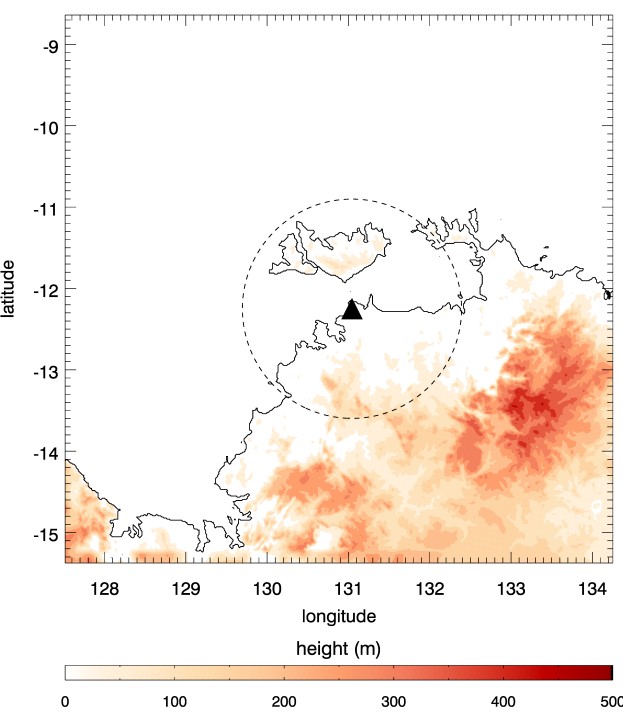

**Figure 2.** Domain for Australian case studies over Darwin showing the Top End of Australia's Northern Territory (which includes Darwin) and the Tiwi Islands. The CPOL radar location is denoted by the black triangle and its coverage by the area within the circle of dashed lines.



**Figure 3.** 18th November 2016 12Z. T+60 Snow amount (kgm-2) (top panels) and Screen temperature (bottom panels) for RAL1-M (left) and RAL1-M with improvements to the Treatment of Lying Snow (right).





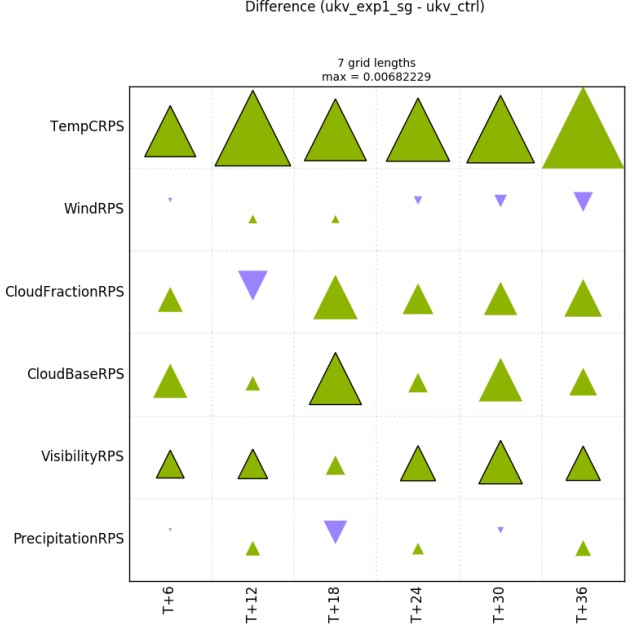

**Figure 4.** HiRA summary scorecard at 10.5km (7 grid-lengths) spatial scale for case studies run with improvements to the Treatment of Lying Snow. HiRA uses synoptic observations (see section 3).

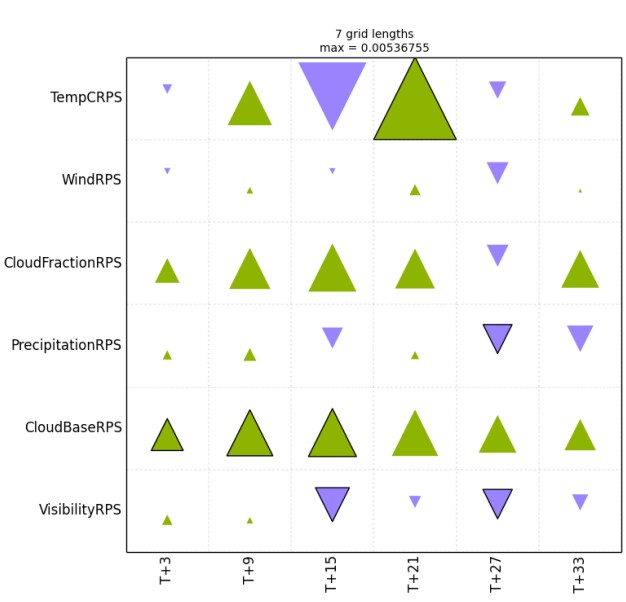

**Figure 5.** HiRA summary scorecard at 10.5km (7 grid-lengths) spatial scale for case studies run with Leonard Terms.





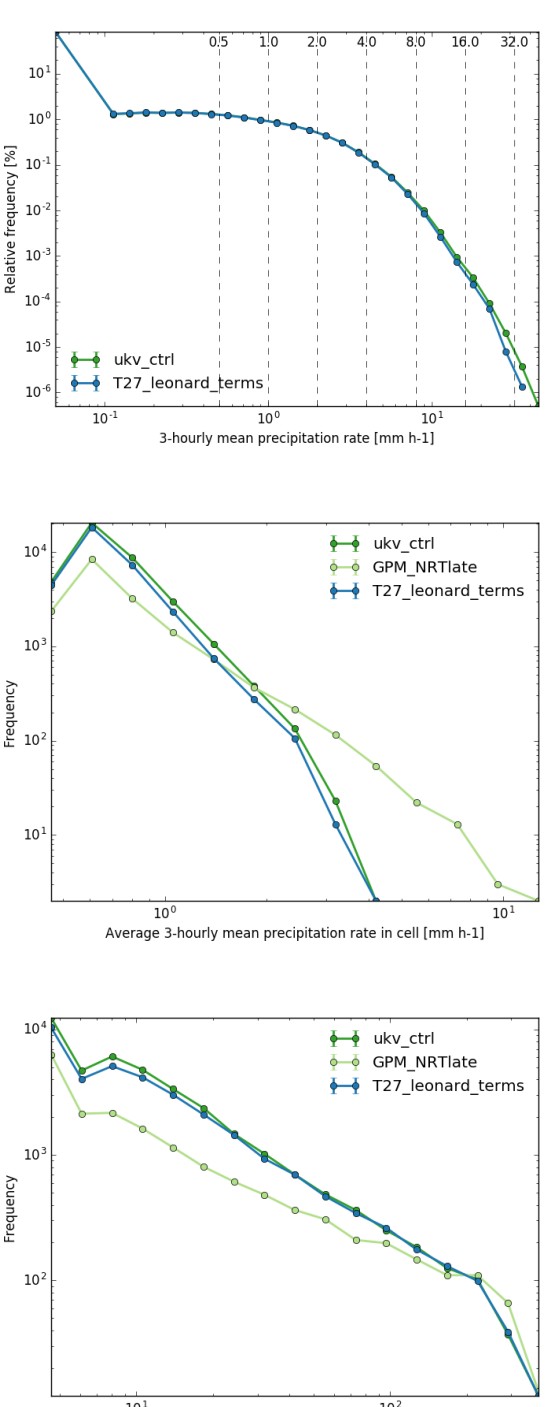

**Figure 6.** 3 hour mean precipitation histogram (top), cell mean value (middle) and cell effective radius (bottom) against GPM NRTlate observations for case studies run with Leonard Terms.



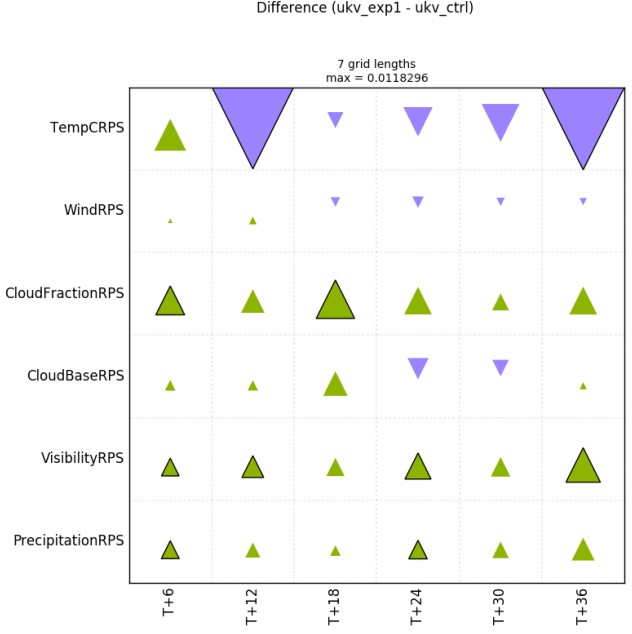

**Figure 7.** HiRA summary scorecard at 10.5km (7 grid-lengths) spatial scale for case studies run with improved ice cloud fraction in mixed phase clouds. HiRA uses synoptic observations (see section 3).

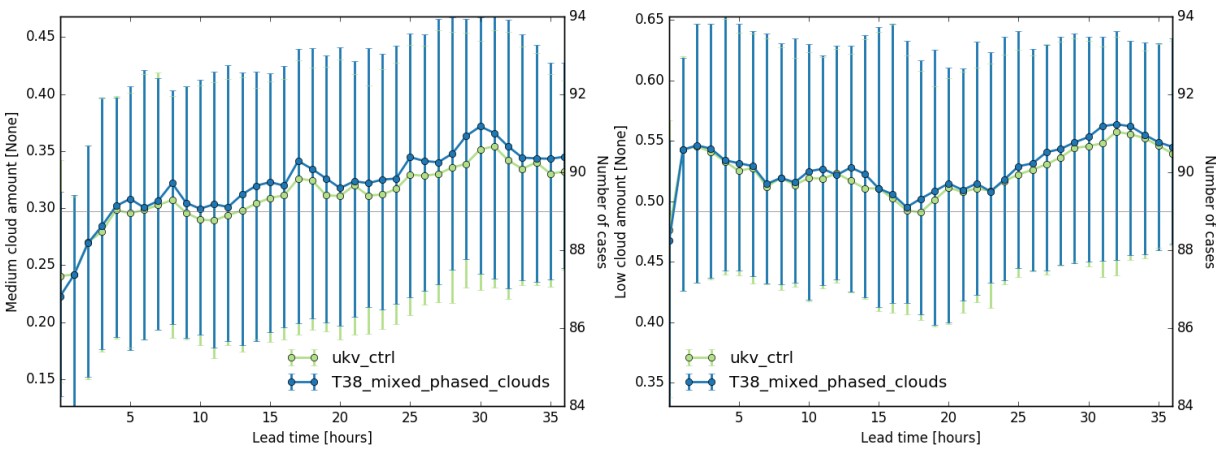

**Figure 8.** Medium cloud (left) and low cloud (right) amounts in case studies run with improved ice cloud fraction in mixed phase clouds.





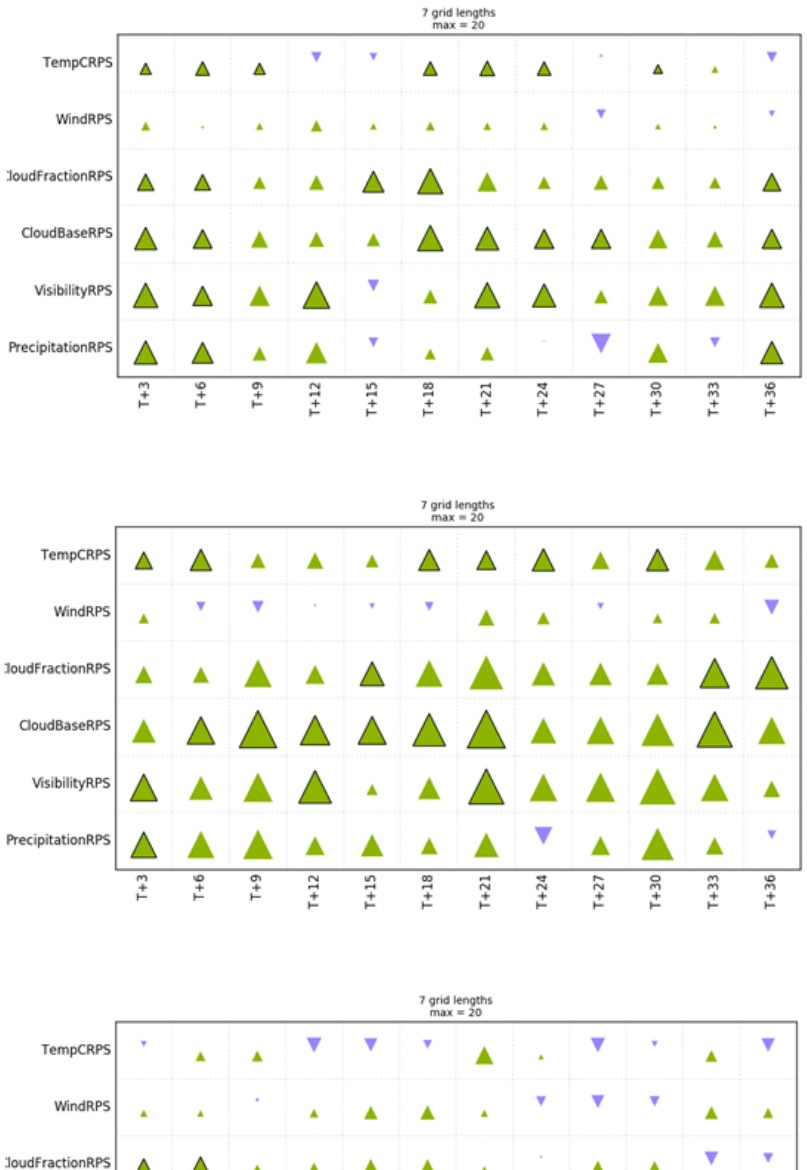

**Figure 9.** Case studies: RAL2-M vs RAL1-M HiRA summary scorecard at 10.5km (7 grid-lengths) spatial scale. Top panel shows results for all cases. The seasonal dependence is explored by stratifying the cases into Winter cases (middle panel) and Summer cases (bottom panel).



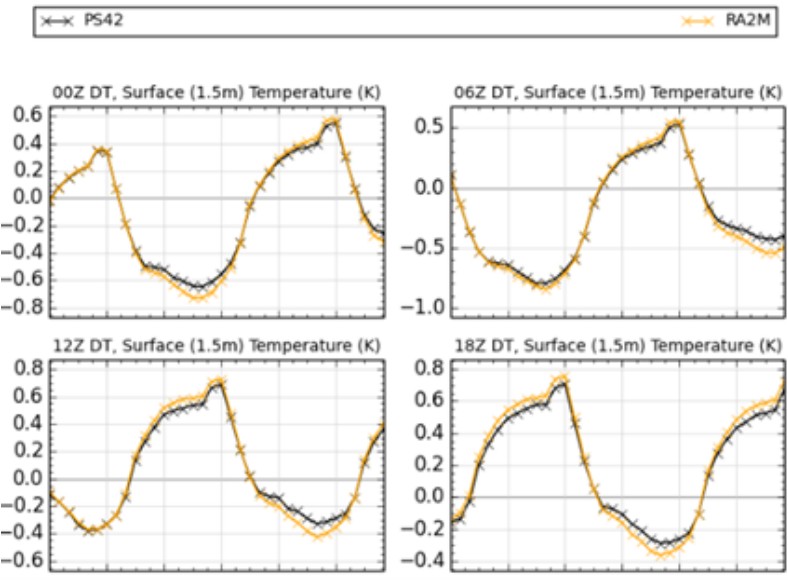

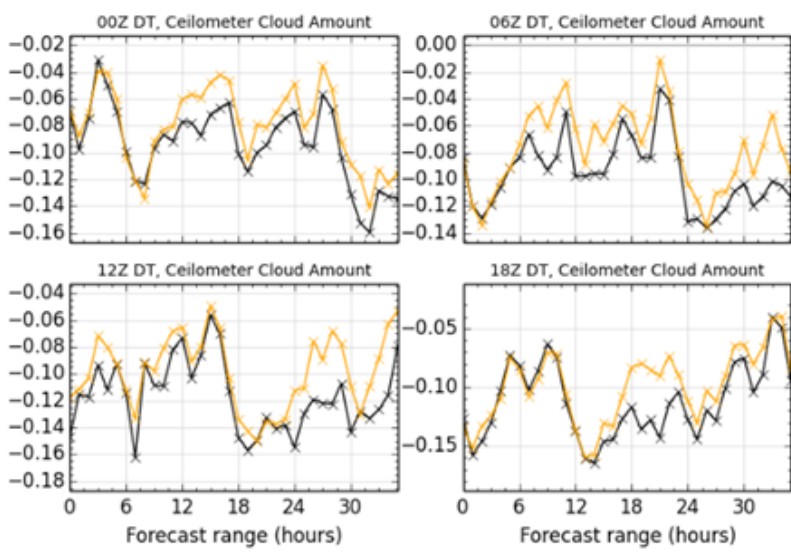

**Figure 10.** 4DVAR trials: Diurnal cycle of Screen Temperature bias (top) and cloud bias (against ceilometer cloud obs)(bottom) in Summer for RAL1 (black) and RAL2 (orange).



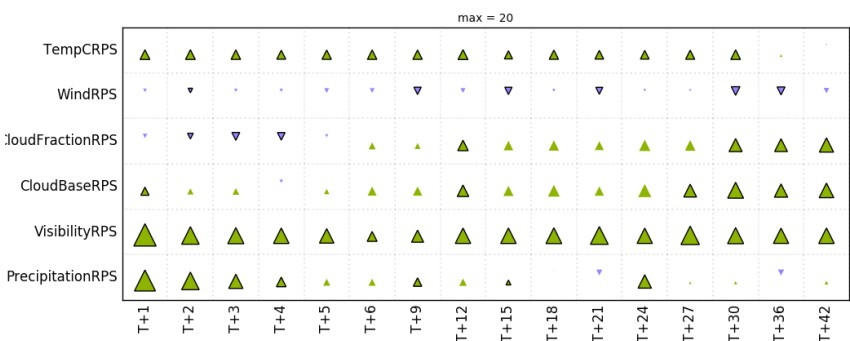

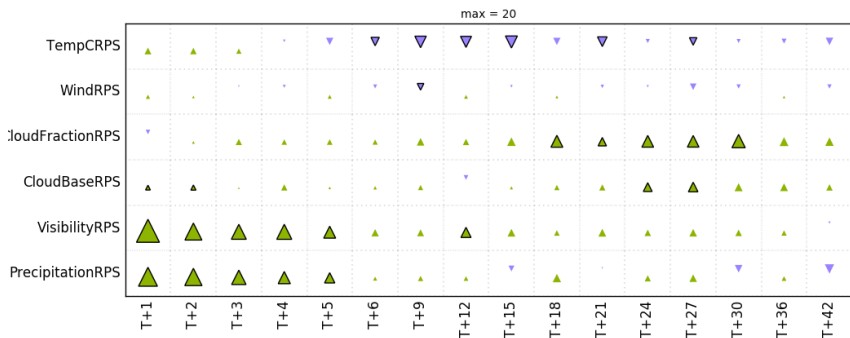

**Figure 11.** MOGREPS-UK trials: RAL2-M vs RAL1-M HiRA summary scorecard at 15km (7 grid-lengths) spatial scale for Winter (top) and Summer (bottom).



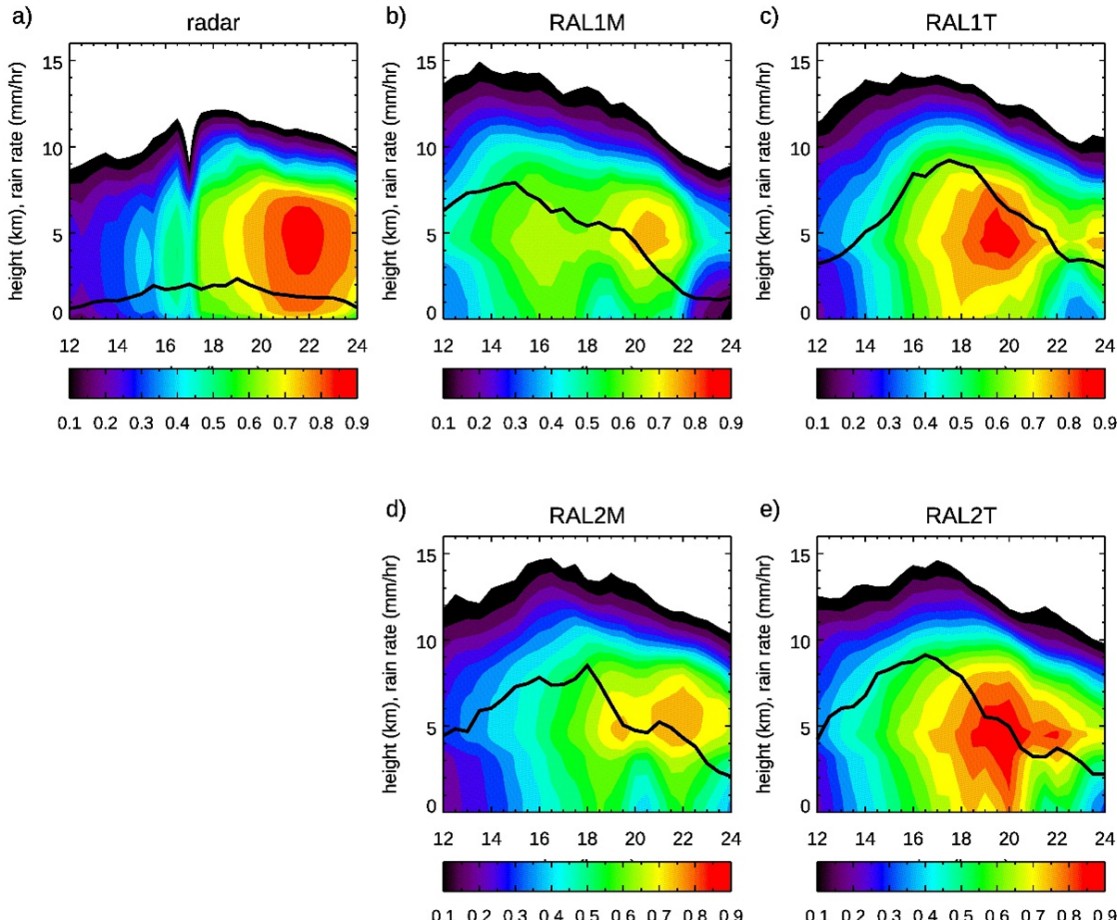

**Figure 12.** Fraction of radar area covered by reflectivities greater than 10 dBZ as a function of height and time (coloured contours) from 12:00 to 24:00 UTC on 18 February 2014. Solid lines are the time series of the domain mean rain rate (mm per hour).





Difference (pcra2_con - ra1t)

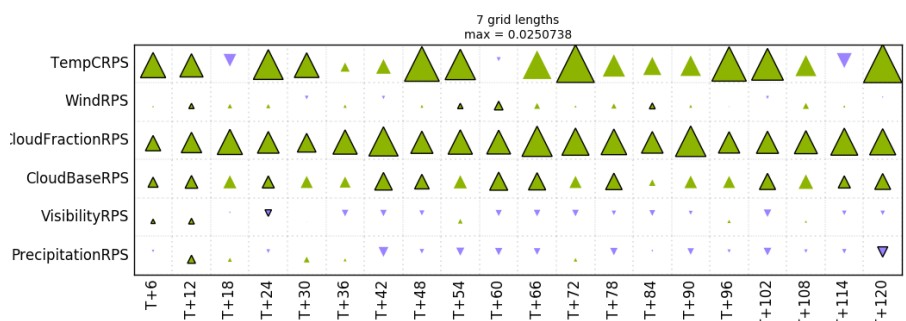

6hr Precipitation Accumulation (mm), Fractions Skill Score (Forecast - Analysis), Area 999,
Equalized, 20181120 00:00 to 20181217 18:00, Unspecified truthtype,
Difference (pcra2_con - ra1t)

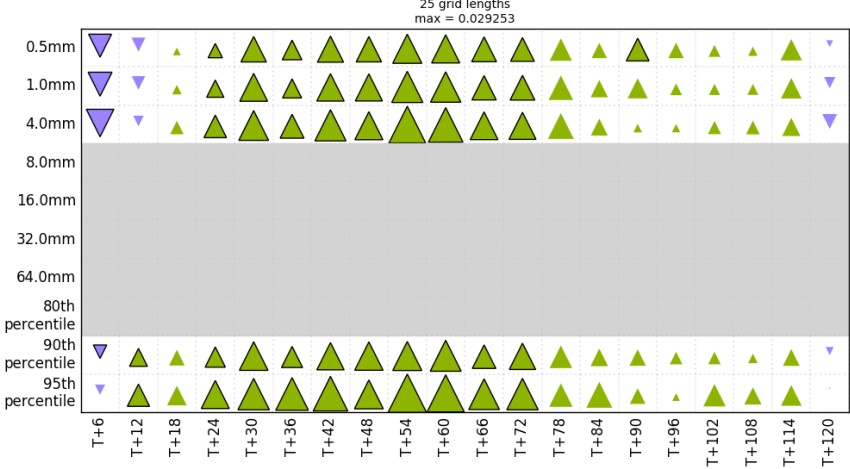

**Figure 13.** RAL2-T vs RAL1-T HiRA summary scorecard at 30km (7 grid-lengths) spatial scale (top) and FSS summary scorecard at 110km (25 gridlengths) for precipitation (bottom).



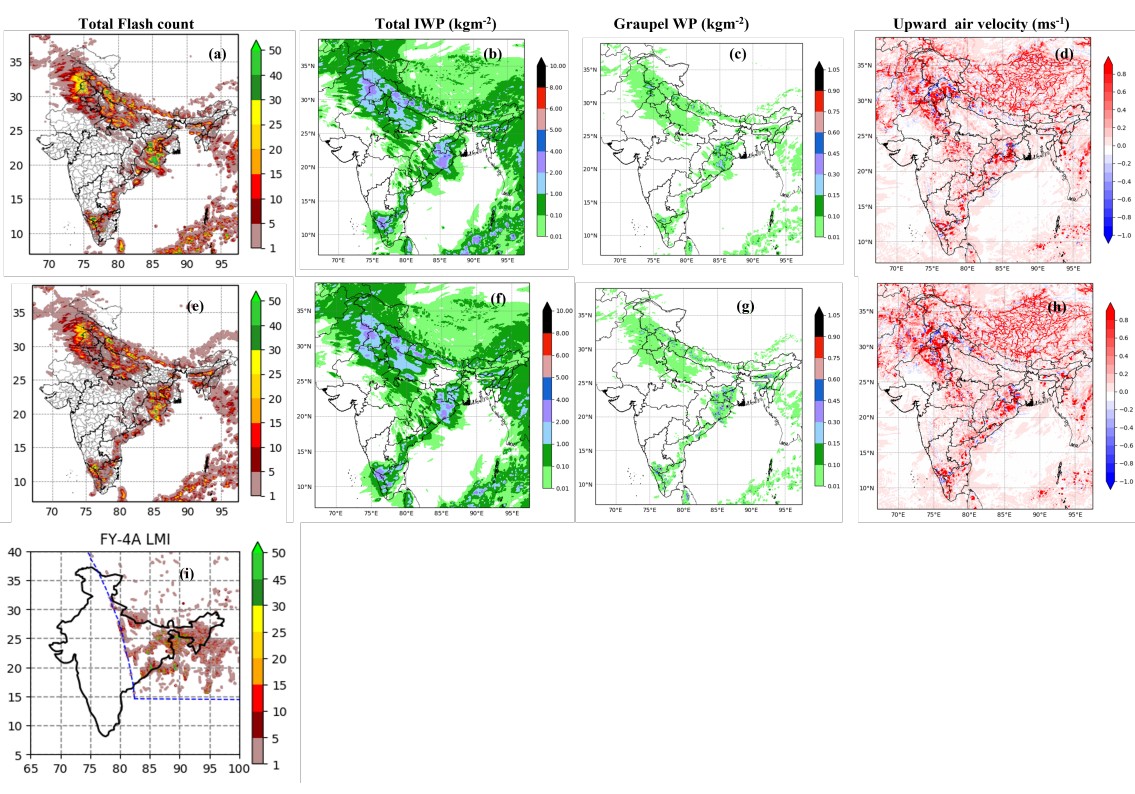

**Figure 14.** NCMRWF 02nd May 2018 lightning case study: Total lightning flashes, Total Ice water Path, Total Graupel Water Path and vertical velocity at 500hPa for RAL1 (top panels) and RAL2 (middle panels). Accumulated lightning flash counts from FY-4A LMI observations (bottom panel).



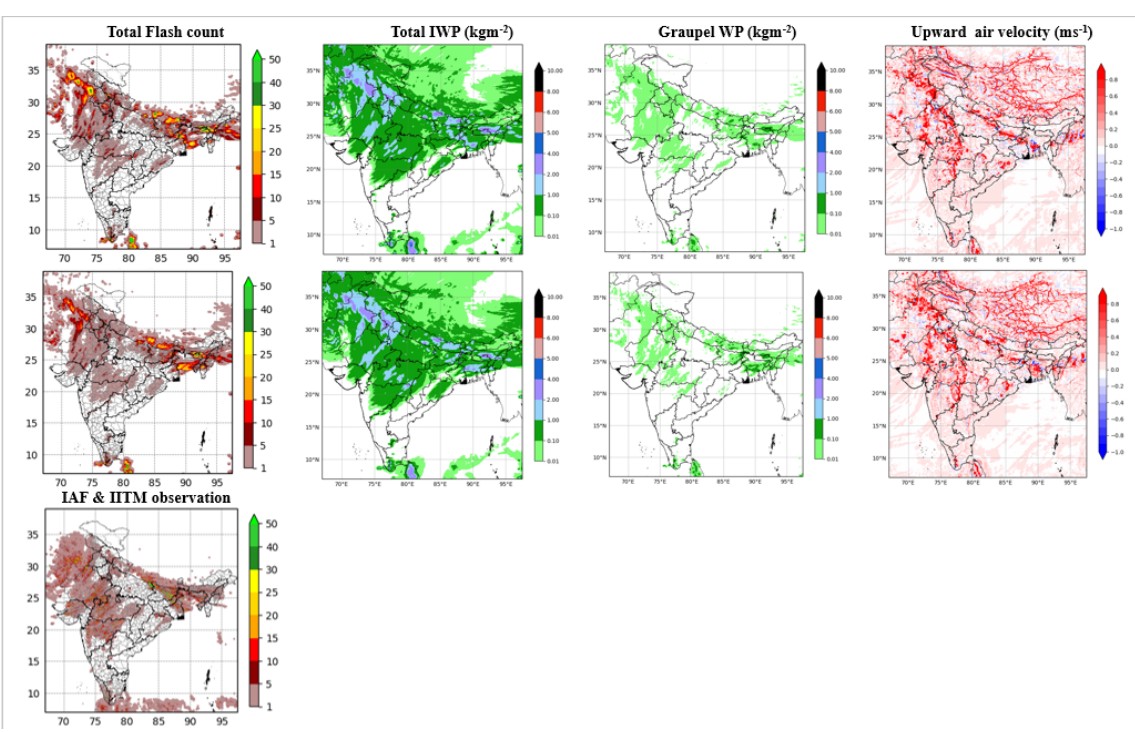

**Figure 15.** NCMRWF 16th April 2019 lightning case study Total lightning flashes, Total Ice water Path, Total Graupel Water Path and vertical velocity at 500hPa for RAL1 (top panels) and RAL2 (middle panels). Accumulated lightning flash counts from IAF and IITM observations (bottom panel).