# Peer review of "The second Met Office Unified Model/JULES Regional Atmosphere and Land configuration, RAL2"

_Geoscientific Model Development, 2022_

## Author Response (AR1)

**Response to RC1 (in bold)**

**Specific comments**

1. p.4, lines 20-27 – The overlap between liquid water and ice phases is "limited" but you don't describe how. Is there a cap on the percent overlap? Is this documented elsewhere and could be referenced?

   **See Appendix of Abel et al. (2017) for more details of the modification to the cloud scheme.**

2. p.5, lines 12-14 – since you show results for addition of the Leonard terms in Figs. 5 and 6, it would be good to give more explanation here of what the Leonard terms actually refer to.

   **The "Leonard term" is an extra subgrid vertical flux that accounts for the tilting of horizontal flux into the vertical by horizontal gradients in vertical velocity. Hanley et al. (2019) found that including this extra term in the Met Office UKV model reduces the peak vertical velocity within updrafts, leading to a reduction in condensation. As a result, the number of grid points with moderate to high rainfall rates, which are overrepresented by the UKV, are also reduced.**

3. Table 2 – what are $b_{LEM}$ and $c_{LEM}$?

   **Replaced Stability function definitions involving bLEM and cLEM in Table2 with "conventional" and "standard". Text now reads: There are two differences in the representation of turbulence between RAL2-M and RAL2-T, namely in the form of the unstable stability functions and in the free-atmospheric mixing length. Both give enhanced turbulent mixing in RAL2-T compared to RAL2-M. RAL2-M uses the Brown (1999) "conventional" function, the same as GA7, while RAL2-T uses the Brown (1999) "standard" function.**

4. p.6, line 12 – seems odd to have "(no reference)" – consider omitting **omitted**

5. p.6, eq.(1) – what is Γ? **Equation deleted and text rewritten as there is a very small impact in RAL2. We don't have a convection scheme, so the parametrized downdraught gustiness velocity scale, w_c is zero (and so gamma is irrelevant). Text now reads: For lower mean wind speeds, the effect of subgrid convective boundary layer gusts on the surface turbulent fluxes is included via a term proportional to the convective velocity scale in the calculation of the friction velocity. For RAL2 we reduce the strength of that term by a half, to then match GL7.0.**

6. p.6, lines 24-25 – why does fixing the multilayer snow scheme allow reintroduction of graupel? Does it form on the snow surface?

   **Improvements to the treatment of lying snow in RAL2 are achieved by introducing a representation of melting of the snow pack from the base over warm ground, as the original code in JULES allows melting only from the surface. Previously it was necessary to remove graupel from the precipitation reaching the surface as the omission of melting from the base resulted in unrealistically prolonged retention of thin layers of frozen precipitation. This modification allows the reintroduction of graupel into the precipitation reaching the surface.**

7. p.7, bottom – the scorecards in the figures use 10.5 km scale (7 grid-lengths). It would be good to explain why this particular scale was chosen for evaluation.

   **This particular scale was chosen for evaluation as Mittermaier and Csima (2017) showed that all variables benefited from the use of at least a 3 x 3 neighbourhood, whilst neighbourhoods which are too large may be detrimental for some variables, including temperature.**

8. p.7, line 18 – define GPM

   **Global Precipitation Measurement (GPM) IMERG satellite data based product (Huffman, 2015, 2017, Skofronick-Jackson et al., 2017) is used.**

9. p.10, line 1 – this should be "the improvement in performance in Winter is much better than the improvement in performance in Summer". **Done**

10. p.10, lines 13-14 and Fig. 12 – do the model results in Fig. 12 correspond to the large rectangular domain over Darwin or for the circular domain of the radar? Please clarify, including in the caption for Fig. 2. If the model results are for the larger domain, how much difference from the radar could be explained by sampling different areas?

    **Figure 2 caption now reads: The CPOL radar location is denoted by the black triangle and its coverage by the area within the circle of dashed lines, which is the area used for the analysis presented in Figure 12.**

11. p.11, lines 4, 9, 10, 14 – define GWP, LMI, ENLS, QCF

    **ENLS : Earth Networks Lightning Sensor**
    **QCF: Cloud Frozen ice content**
    **GWP: Graupel Water Path**
    **LMI : Lightning Mapping Imager**

12. p.11, lines 17-18 – readers may not know where these regions are – it would be better to say what part of India (northern, etc.)

    **Bihar and Uttar Pradesh (Eastern India) and also very few flash strikes over the Rajasthan-Madhya Pradesh border (north-west India).**

13. p.12, lines 6-10 – do you think the implementation of RAL2 in operations with only 70 levels rather than 90 levels (as shown in the results in this paper) has much effect on the improvements over RAL1? If this has been tested it would be good to say a bit more about it.

    **Both level sets have a very similar number of near-surface levels with both having 28 levels below 3km and L90(67t,23s)40 only having one extra level by 10km asl. As a consequence of this, the impact of 10 L90(67t,23s)40 was found to be very small for this UK specific application.**

**14.** p.A table of acronyms is provided in Appendix 2 but never referred to in the text.

**Now referenced at the beginning of chapter 2**

**Technical corrections**
1. p.1, line 18 and elsewhere – There are too many parentheses in the in-line citations. This should be (e.g. Baldauf et al., 2011; Brousseau et al., 2016; Bengtsson et al., 2017; Klasa et al., 2018).
   **\cite[e.g.][]{Baldauf:2011,Brousseau:2016,Bengtsson:2017,Klasa:2018}**
2. p.2, line 6 – Fix spelling of 'the' **removed double the**
3. p.2, line 16 – RAL has not yet been defined in the body of the paper - **defined**
4. p.2, lines 30, 32 and elsewhere – Instead of "Sect" write "Section" **done**
5. p.4, line 10 – fix "but extensively modified is used" **reworded**
6. p.4, line 20 – do you mean "a change in RAL2-M"? **Changed at to in**
7. p.5, line 21 – change "RAL2 to use" to "RAL2 uses" **done**
8. Tables – remove "%" from captions   **removed {$\%$}}**
9. p.9, line 6 and 8, and elsewhere in case studies – remove the "0" from "04th December 2019" **done**
10. Results sections – no need to capitalise Winter and Summer **done**
11. p.9, line 23 – should be "Figure 11 shows RAL2 outperforms RAL1…" **done**
12. p.10, line 20 – rather than "in the longer time" give the hours for which RAL2-T performance is better than RAL1-T

**Response to RC2 (in bold)**

**\*Specific Comments\***
Where case studies and trials of the complete RAL2 configuration are described for the UK (Sec 3.3, 3.4, 3.5) these have been related back to the one or more of the individual changes (i.e. the performance described in Sec 3.2). However, the other cases are not so well linked. For example:
a) in the MCS case it is not clear whether the improvement in fractional coverage shown in Fig 12 is to be expected, nor which of the science changes might have caused this.

**The increase in fractional coverage of cloud and rain in the MCS case is likely to be predominately a result of the introduction of the Leonard terms (ticket 27). This science change has the impact of reducing the number of small convective cells and heavy rain rates (Fig.6) and helps to produce better cloud cover, as indicated in Figure 5. However, to be able to quantify the contribution of each science change to the change in, for example cloud cover, would be require many more simulations that include both individual and combinations of changes, which is outside the scope of this study. Therefore, we propose that the text remains unchanged as we don't have strong evidence to confidently address this comment.**

b) in the South East Asia cases, what might be causing the degradation in FSS during spinup, or rather, why does RAL2-T take longer to spinup than RAL1-T. Is it due to the BL stochastic perturbations (Table 2) or was there no change in this from RAL1 to RAL2?

**There was no change to the BL stochastic perturbations between RAL1 and RAL2. The change is likely to be predominately a result of the introduction of the Leonard terms (ticket 27). The spin-up with RAL2-T is more muted than with RAL1-T. The reduced rainfall amounts means that the absolute thresholds are worse (not significantly though), but the percentile thresholds are significantly better.**

c) for the Indian lightning cases, I assume the changes seen here are partly/mainly?? due to the liquid and ice phases in the cloud scheme (Sec 2.6).

**The change to limit the overlap between the liquid water and ice phases was only applied to RAL2-M and is not applicable to RAL2-T.**

There is mention of a reduction in graupel and ice water paths, but it's not obvious what is causing this. Also, I assume the results shown are after the reduction in GWP threshold was applied to the RAL2-M configuration; if so it would be helpful to see what the results from the 'standard' configuration looked like.

**The GWP threshold reduction was not applied to RAL2-M. All experiments were performed using RAL2-T.**

This threshold adjustment seems arbitrary, why tune RAL2 to RAL1 output? From Fig 14 & 15, it looks like both RAL1 and RAL2 are producing higher flash counts than the obs (albeit with lower spatial coverage) so its not clear why they should be increased.

The threshold adjustment to RAL2-T was made to increase the area coverage of the moderate lighting flash counts distribution which was for many cases even less than observations at many locations. By reducing the threshold, not only the area coverage, but the intensity also increases at some locations.

Agree that both RAL1-T and RAL2-T are producing higher flash counts compared to observations after the tuning of RAL2-T. Our strategy is to reduce the missing events at the cost of some false alarms. However, we have conducted experiments with many lightning events of light, moderate and extreme intensity, and the overall objective scores (Reply-to-reviewer Table.1) are favoring RAL2-T compared to RAL1-T. Hence definitely there is an improvement over RAL1-T.

Reply-to-reviewer Table.1 Objective scores of daily accumulated lightning for RA1T and RA2T (8-12 August 2019) over Indian domain (Root mean Square error, Correlation coefficient, Bias, Multiplicative bias, Mean forecast, Mean observation).

| Statistics | LIGHTNING | |
|---|---|---|
| | RA1T | RA2T |
| RMSE | 8.745589 | 8.336644 |
| Corr. Coef. | 0.032567 | 0.039343 |
| Bias | 0.452464 | 0.299027 |
| Mult.Bias | 1.673237 | 1.495751 |
| Mean Fcst | 1.46396 | 1.310521 |
| Mean Obs | 1.011495 | 1.011495 |

*Technical Corrections*
(Apologies, but some of these get rather pedantic...)
Pg2, Ln 5: The 'regional model' you refer to here has not been defined yet (except in the abstract). It needs to be defined as UM/JULES or similar. (Also 'trhe world' sp.)
**Moved definitions further up in the Introduction.**
Pg2, Para 3/4: I think it would make sense to introduce RMED at this point (rather than just as a prefix when introducing the toolbox). Or if not here, then around line 7 on page 7 where the regional model evaluation process is introduced.
**Introduced around line 7 on page 7. The Regional Model Evaluation and Development (RMED) team at the Met Office carry out scientific research and technical developments to improve current and next-generation regional modelling systems. RMED develops and delivers regional model configurations (e.g RAL2) for use in weather forecasting and climate prediction; develops tools and methods for effective model evaluation; evaluates and develops next generation convective scale models and builds, tests and evaluates the science of coupled regional modelling systems.**
Pg3, Ln2: When introducing section 2 I think it is implied that when not stated explicitly, then the definitions given haven't changed from RAL1 to RAL2 (e.g. Sec 2.2, 2.3, 2.4). It would be helpful to make this explicit.

**Certain aspects of the model (e.g those described in sections 2.2, 2.3, 2.4, 2.5 and 2.9) haven't changed from RAL1 to RAL2.**

Pg3, Ln6: The SOCRATES url in the footnote is not accessible (I assume this is an internal website). Remove. **Removed**

Pg3, Ln 19: Delete 'basic underlying'. **Removed**

Pg6, Ln25: Thin snow albedo bugfix: this should either be expanded on or removed. Did the bug have a noticeable effect? **Removed**

Pg6, Ln26: Similarly the snow grain growth. Is there a reference for this? Why was the treatment revised? **Removed**

Pg8, Ln4: Delete "simple" **Removed**

Pg8, Sec 3.2: Can you confirm that all the scorecards and results (Fig 4, 5, 6, 7 & 8) are based on all 100 UK cases? And do they include the Darwin cases? How many Darwin cases are included?
**No Darwin cases are included in these figures. Figure 3 to Figure 8 show results from these 100 cases, which were downscaling runs (from the Met Office Global model) with no data assimilation.**

Pg9, Ln1: Clarify that "performance" in winter is not necessarily better than summer. "... by season reveals that the IMPROVEMENT IN performance in winter is LARGER than in summer..." **Stratifying the cases by season reveals that the improvement in performance in winter, where almost all parameters are improved (Figure 9 middle panel), is much greater than the improvement in performance in summer (Figure 9 bottom panel).**

Pg9, Ln6-8: "04th" shouldn't have the "0" **Removed**

Pg10, Ln7: I think "though" is meant to be "through" **Corrected**

Pg11, Ln22/Ln25: "RAL2 Science" and "Data Assimilation" don't need capitals **Corrected**

Pg11, Ln25: "(the latter we refer to as case studies)" This is probably not necessary. But if it is important should be pointed out in the relevant earlier section, not in the conclusion.
**Results are presented from case studies with domains in both the mid-latitudes (U.K and Perth in Australia) and the tropics (Darwin in Australia, South East Asia and India)**

Pg12, Ln7: change "RAL2-M science is running 24/7 in" to "where RAL2-M science is used in" **changed**

Pg12, Ln6-8: "Operationally" doesn't need capitals and '04th' should be '4th' **Corrected**

Pg12, Ln13: change "the partnership" to "the UM Partnership" **Corrected**

Acronyms not defined:

Pg2, Ln4: UM (except in abstract where an acronym isn't needed, should be defined in main text)

Pg6, Ln20: LES & PBL **deleted**

Pg7, Ln18: GPM **defined**

Pg11, Ln4: GWP **defined**

Seasons shouldn't be capitalised (e.g. winter and summer throughout page 9, and Fig 9-11 captions) **Corrected**

Captions to Fig 3-6 have unnecessary capitals. I don't think they need to match the 'Description' field in table A3. Also too many capitals in caption of Fig 14. **Corrected**

**Response to RC3 (in bold)**

**Specific Comments:**
To fully understand some references and model configurations, one needs to have a good understanding and knowledge of the contents and results of the 2020 paper. Therefore, some technical references or specific comments might be seen as unnecessary.

1. During the explanation of rotated v. unrotated grids, it is mentioned that Australia is at lower latitudes (p 3 line 7). However, the continent lies between 10-43 degrees south. Therefore, it might be worthwhile to give a short explanation/reasoning why regional domains for parts of Australia does/does not apply a rotated grid. Alternatively, since the domain is not yet defined y p3, mention that the domain of interest in this paper lies within the tropics; hence the model is unrotated.
**In contrast, domains which lie within the tropics use unrotated grids and this applies to the domains of interest in this paper over Darwin, Australia, South East Asia and India**

2. Remove "%" in all table headings **done**

3. P2, L13: Do forecasters also conduct subjective assessments at NCHWRF and BoM, or is this done only at the Met Office? **This paragraph is about the Met Office practices. NCMRWF also conducted objective and subjective assessments of daily rainfall and lightning flash counts with a few extreme events as well as moderate events, though not extensively with other variables.**

4. P5, Table 2: Define $b_{LEM}$ and $c_{LEM}$. **Replaced Stability function definitions involving bLEM and cLEM in Table2 with "conventional" and "standard". Text now reads: There are two differences in the representation of turbulence between RAL2-M and RAL2-T, namely in the form of the unstable stability functions and in the free-atmospheric mixing length. Both give enhanced turbulent mixing in RAL2-T compared to RAL2-M. RAL2-M uses the Brown (1999) "conventional" function, the same as GA7, while RAL2-T uses the Brown (1999) "standard" function.**

5. P6 Equation 1: Define $\Gamma$. **Equation deleted and text rewritten as there is a very small impact in RAL2. We don't have a convection scheme, so the parametrized downdraught gustiness velocity scale, w_c is zero (and so gamma is irrelevant). Text now reads: For lower mean wind speeds, the effect of subgrid convective boundary layer gusts on the surface turbulent fluxes is included via a term proportional to the convective velocity scale in the calculation of the friction velocity. For RAL2 we reduce the strength of that term by a half, to then match GL7.0.**

6. P6,L24: Briefly explain on how/why changes in surface snow settings affect graupel in precipitation. **Improvements to the treatment of lying snow in RAL2 are achieved by introducing a representation of melting of the snow pack from the base over warm ground, as the original code in JULES allows melting only from the surface. Previously it was necessary to remove graupel from the precipitation reaching the surface as the omission of melting from the base resulted in unrealistically prolonged retention of thin layers of frozen precipitation. This modification allows the reintroduction of graupel into the precipitation reaching the surface.**

7. P7,L20: Explain why scorecards use 10.5 km (7 grid-lenghts); possibly related to the scale of the synoptic observations? **This particular scale was chosen for evaluation as Mittermaier and Csima (2017) showed that all variables benefited from the**

**use of at least a 3 x 3 neighbourhood, whilst neighbourhoods which are too large may be detrimental for some variables, including temperature.**

8. In Section 3.2: It is overall not clear which case studies are evaluated and presented in Figures 4-8. It is stated that RAL1-M were evaluated for UKV and Darwin, but not the number of cases for Darwin. **Individual science changes (see list of RMED tickets in Table A3) were tested by running 100 case studies with a 1.5km horizontal grid-length, using the same domain as the Operational UKV model (Figure 1). Figures 3 to 8 show results from these 100 cases, which were downscaling runs (from the Met Office Global model) with no data assimilation.**

- Which case studies are included in the results shown in the scorecards, and how many cases from each partner were included? Apart from the 100 cases at the Met Office, it is not stated how many cases were conducted by NCMWRF and BoM. **No Darwin cases are included in these figures. Figures 3 to 8 show results from these 100 cases, which were downscaling runs (from the Met Office Global model) with no data assimilation.**

- In the scorecards, does "precipitation" include snowfall? **Yes, it does.**

- Figure 6: Does the possible weakness of GPM capturing higher rain rates contribute to this result? Or is this not applicable in higher latitudes?

- Figure 6: Was the model resolution upscaled to the GPM resolution? **Yes. For each cell statistic plot, the necessary model and observational data is read in from the netCDF file regridded_cubes.nc in each model data directory. The netCDF file will contain the model fields (and any corresponding gridded observations) required for each cell statistic plot, on one or more common spatial grids.**

- P8,L26: To which Figure does this paragraph refer? How did you distinguish between Met Office and Darwin cases? **Whilst tickets 20, 27 and 38 had a positive impact over the UK, there was a neutral impact over Darwin (hence no results shown in this section). Likewise no results are shown from tickets 30, 36, 37, 39, 42 and 43 as their impact was neutral, showing no statistically significant changes in performance over either the UK or the Darwin domains.**

9. P9,L1: No plausible explanation is given for the performance increase in winter and not summer months. **Hard to give a definitive reply to this question. Some changes such as #20 will clearly be more active in winter. Also depends on seasonality of biases. The increase in cloud in summer increases a cold daytime bias.**

10. P9,L23: Define the grid resolution of MOGREPS-UK since, in Figure 11, 7 grid-lengths now equal 15 km as opposed to 10.5 km in previous scorecards.

11. P10,L4: Is the more extensive fog in the RAL-M configs more accurate? Also, is it more extensive in both temporal and spatial scales compared to the observed? **It is hard to give a definitive reply to this question. There is no strong signal for a systematic impact of RAL2 changes on fog. There is a much larger variability from case to case.**

12. P10,L13: "Figure 2 shows.." – is the results in 3.7/Figure 12 only for the circle (radar coverage) in Figure 2 or for the whole domain as shown in Figure 2? Please clarify. **Figure 2 caption now reads: The CPOL radar location is denoted by the black triangle and its coverage by the area within the circle of dashed lines, which is the area used for the analysis presented in Figure 12.**

13. P11: Define GWP, LMI, ENLS, QCF
   **ENLS : Earth Networks Lightning Sensor**

**QCF: Cloud Frozen ice content**
**GWP: Graupel Water Path**
**LMI : Lightning Mapping Imager**

14. Table A2 is not referred to in the text **Now referenced at the beginning of chapter 2**

**Technical corrections:**

1. Figures 1 and 2: use either "orography" or "height" in both figures for consistency. **Not done**
2. P1, L18: It is suggested that the term "National Hydrological and Meteorological Services" (NHMS) be used to align with international practice (also used by the WMO). **Corrected**
3. P1, L19: Correct the reference syntax; include all references in one bracket. **Corrected**
4. P2, L6: Typing error "trhe" **done**
5. P2, L8: Suggest shortenening the sentece: "…10 years. This strategy includes…" **The Met Office Research and Innovation Strategy sets out aims for the next 10 years across science, technology and operations. One of the key themes is pulling through science into services and this includes RAL science configurations**.
6. P2, L21: "The systems run in variable…" **done**
7. P2: L23: Suggest defining UK (might be pedantic) **southern part of the United Kingdom**
8. P3,L5: "…centre of the regional model domain…" **done**
9. P3,L10: Swap RAL2 and RAL1 in the sentence to agree with the order as shown in Table 1. **done**
10. P3,L17-8: Add a reference for this statement **Not sure which statement - the first or the second?: "The rationale for these differences is that the tropopause is shallower in the mid-latitudes than in the tropics. Also, boundary layer fog and low cloud processes are more important in the mid-latitudes and convection more important in the tropics".**
11. P4,L6: The link is to an internal site at the Met Office and is inaccessible to the public. Suggest using the University of Leeds site for the PDF document (http://homepages.see.leeds.ac.uk/lecsjed/winscpuse/socrates_techguide.pdf ) website OR the SOCRATES github page (https://execlim.github.io/Isca/modules/socrates.html) similar. **done**
12. P4,L10: Give an example of how the single moment microphysics has been "extensively modified." **Prognostic rain and prognostic graupel are included. The warm-rain scheme is based on Boutle et al. (2014) whilst ice cloud parametrisations use the generic size distribution of Field et al. (2007) and mass-diameter relations of Cotton et al. (2013).**
13. P4,L20: Provide a reference or briefly explain why a limit is applied and the possible ranges between liquid water and ice phases. **(see Appendix of Abel et al. (2017) for more details of the modification to the cloud scheme)**
14. P5,L3: Reference for statement ending with "… kilometre scale models" **(Takayabu et al., 2022)**
15. P5,L5: Suggestion for consistency, use "s" in both parameterisation" and "parameterise", although "z" is an acceptable British spelling. **done**
16. P6,L8: Has GL been defined? **done**

17. P7,L31: Again remove "%" in table title **done**
18. P7,L9: Suggest replacing "variety" with "diverse" to eliminate the use of "variety" in consecutive sentences. **done**
19. P7,L10: Define RMED earlier since it refers not only to the Toolbox but also to the larger collaboration in the UM Partnership on regional model evaluation. **done**
20. P7,L18: Define "GPM".  **Global Precipitation Measurement (GPM)**
21. P8,L20: "observations" and not "obs". **done**
22. P8,L28: Typing error "togther". **done**
23. P9,L1: "reveals that almost all parameters improved during the winter months, as opposed to summer". **done**
24. P9,L8: 4$^{th}$ **done**
25. P9,L6, L11 and L12: Use dates syntax consistently and throughout the rest of the paper. **done**
26. P9,L11: What were the results for the winter months, or was the sample size too small? **Results were consistent with case studies (section 3.3) and MOGREPS-UK trials (Section 3.5)**
27. P10,L24: Suggest changing to "model for a large domain covering …", **done** indicated in Figure 14. **Not Figure 14 which is an NCMRWF figure.**
28. P10,L28: It might be worth mentioning why the results from t+72 onwards are not of significance or not described in the results. **and a smaller improvement out to T+114**
29. P11,L7 and 17: Where are these regional areas located? Possibly show on a map to support the reader in interpreting the results. **over the foothills of Bihar and Uttar Pradesh (Eastern India) and also very few flash strikes over the Rajasthan-Madhya Pradesh border (north-west India).**
30. P11,L26: U.K to UK **done**
31. P11,L19: "The maximum flash counts…" **done**
32. P31: Figure 11, 13 – Reduce the space between the title and graph. **done**
33. P34: Figure 14: Panels a) to i) is not defined or referred to in the text/Figure title. Suggest using the panel references when discussing the results for ease of reference to the maps. Conversely, Figure 15 for the second case study does not indicate panel numbering. **done**